# Diacylglycerol Kinase alpha in X Linked Lymphoproliferative Disease Type 1

**DOI:** 10.3390/ijms22115816

**Published:** 2021-05-29

**Authors:** Suresh Velnati, Sara Centonze, Federico Girivetto, Gianluca Baldanzi

**Affiliations:** 1Department of Translational Medicine, University of Piemonte Orientale, 28100 Novara, Italy; suresh.velnati@med.uniupo.it (S.V.); sara.centonze@uniupo.it (S.C.); 20020549@studenti.uniupo.it (F.G.); 2Center for Translational Research on Allergic and Autoimmune Diseases (CAAD), University of Piemonte Orientale, 28100 Novara, Italy

**Keywords:** signal transduction, activation-induced cell death, PKC, ERK, SHP-2, SLAM, SH2D1A

## Abstract

Diacylglycerol kinases are intracellular enzymes that control the balance between the secondary messengers diacylglycerol and phosphatidic acid. DGKα and DGKζ are the prominent isoforms that restrain the intensity of T cell receptor signalling by metabolizing PLCγ generated diacylglycerol. Thus, their activity must be tightly controlled to grant cellular homeostasis and refine immune responses. DGKα is specifically inhibited by strong T cell activating signals to allow for full diacylglycerol signalling which mediates T cell response. In X-linked lymphoproliferative disease 1, deficiency of the adaptor protein SAP results in altered T cell receptor signalling, due in part to persistent DGKα activity. This activity constrains diacylglycerol levels, attenuating downstream pathways such as PKCθ and Ras/MAPK and decreasing T cell restimulation induced cell death. This is a form of apoptosis triggered by prolonged T cell activation that is indeed defective in CD8^+^ cells of X-linked lymphoproliferative disease type 1 patients. Accordingly, inhibition or downregulation of DGKα activity restores in vitro a correct diacylglycerol dependent signal transduction, cytokines production and restimulation induced apoptosis. In animal disease models, DGKα inhibitors limit CD8^+^ expansion and immune-mediated tissue damage, suggesting the possibility of using inhibitors of diacylglycerol kinase as a new therapeutic approach.

The investigation of the molecular mechanisms underlying X linked proliferative disease type I (XLP-1) have evidenced a reduced intensity of T cell receptor (TCR) signalling strength [1] and a peculiar defect in diacylglycerol (DAG) mediated signalling [2]. The shreds of evidence indicating an involvement of diacylglycerol kinase α (DGKα) in this phenotype are presented in here together with a possible implication for the design of targeted XLP-1 therapies.

## 1. Introduction

DAG is a key second messenger in T cell physiology that promotes membrane recruitment and activation of several effectors. DAG activates conventional and novel protein kinase C (PKC) along with Ras guanine-releasing protein-1 (RasGRP1) and other C1 domain-containing signal transducers [3]. In T cells the majority of receptor-induced DAG is produced by the action of phospholipase C γ1 (PLCγ1) on membrane phosphatidylinositol 4,5 bisphosphate. PLCγ1 is crucial for T cell activation in terms of proliferation and cytokine secretion [4] by acting upstream to kinases such PKC and the mitogen-activated protein kinase cascade (MAPK) and also of key transcription factors such as nuclear factor of activated T-cells (NFAT), nuclear factor-kappa light chain enhancer of activated B cells (NFκB) and activator protein 1 (AP1) [5]. In particular, DAG at the plasma membrane starts the MAPK pathway by bringing RasGRP1 close to Ras [6,7] and at the same time it activates conventional and novel PKCs by abrogating the pseudo-substrate binding to the catalytic domain [8]. Both DAG dependent pathways are necessary for immune synapse organization and full T cell activation [9]. Interestingly, T cell activation in absence of costimulatory signals drives those cells in anergy. This is a hyporesponsive status that contributes to peripheral immunotolerance, characterized by reduced Ras signalling due to DGKα overexpression, resulting in defects in lymphocyte proliferation and IL-2 production [10]. In line with a modulatory role of DAG metabolism, DGKα inhibitors not only rescue anergic cells but also reinvigorate exhausted tumour infiltrating lymphocytes, suggesting that this isoform plays a key role in the negative regulation of T cell effector functions [11]. 

The regulation of DAG levels in T cells is the result of a balance between the synthesis by PLCγ1 and the metabolism mediated by DGK as evidenced by the hyperresponsive phenotypes of DGKα and DGKζ deficient lymphocytes [12,13]. DGKα and DGKζ are both involved in the negative control of TCR signalling with some differences: DGKζ appears to play a quantitatively predominant role at the plasma membrane, while DGKα has a specific role in shaping the DAG gradient at the immune synapse [14]. Blocking DGKα or DGKζ activity potentiates TCR signalling along with the MAPK/AP-1 axis and NFκB activity, resulting in enhanced expression of T cells activation markers such as CD69 and Nur77 [15,16]. 

## 2. X-Linked Lymphoproliferative Disease Type 1

XLP-1 is a rare form of primary immunodeficiency affecting about one-two out of one million males, resulting in an increased vulnerability to Epstein-Barr viral (EBV) infection. Although the exposure of patients with XLP-1 to EBV induces an uncontrolled immune response including the activation of lymphocytes and monocytes, this response is not able to eradicate the infection [17]. Moreover, EBV persistency may evolve in severe manifestations such as hemophagocytic lymphohistiocytosis (HLH). While HLH is almost always caused by EBV infection, other manifestations are present in XLP-1 EBV^-^ patients such as malignant lymphoma, hypogammaglobulinemia or dysgammaglobulinemia, bone marrow hypoplasia and lymphocytic vasculitis. This suggests that the exposure to EBV is not responsible for all the clinical features of the disease [18,19,20].

Mutations in XLP-1 are localized to the SH2D1A gene, a small 4-exon gene located in the long arm of chromosome X (Xq25). SH2D1A encodes for a 128 aa protein named signalling lymphocyte activation molecule (SLAM)–associated protein (SAP). SAP is an adaptor protein consisting of an N-terminal domain of five amino acids, a central SH2 domain of approximately 100 amino acids and a C-terminal region of nearly 20 amino acids [21,22]**.** SAP is expressed in T cells, natural killer (NK), and invariant NKT (iNKT) cells. According to Sayos and colleagues**,** SAP expression is detectable in the majority of human T cells subsets (CD4^+^, CD45RO^+^, CD45RA^+^ and CD8^+^), in the T-cell tumour cell line Jurkat, and in the Burkitt lymphoma line Raji which is positive in the EBV [22].

The SH2D1A protein product binds to the cytoplasmic portion of the SLAM family of transmembrane receptors. Binding occurs between a conserved Immunoreceptor Tyrosine-based Switch Motif (ITSM) in SLAM and the SAP SH2 domain in which the arginine residue at position 32 has been shown to play a critical role [22]. Studies of crystallographic and nuclear magnetic resonance reported that the interaction between the SAP SH2 domain and its ligand occurs through an atypical binding mode [23,24]. Generally, SH2 domains bind to their ligands by a ‘two-pronged’ mechanism, where they simultaneously interact with the phosphorylated tyrosine and the related C-terminal residue. When it turns to SAP, the interface with its ligand (SLAM) involves an additional interaction between the SH2 domain and residues located N-terminal to the phosphotyrosine, generating a ‘three-pronged’ association. The presence of this third interaction considerably potentiates the affinity of the SH2 domain of SAP for SLAM (*K*_d_≈120–150 nm; compared to *K*_d_≈500 nm for traditional SH2 domains) [25]. On the other hand, SAP not only binds to SLAM but also works as an adapter protein by interacting with the protein tyrosine kinase Fyn. The interaction between SAP and Fyn involves the arginine residue in position 78 while SAP–SLAM interaction happens at position 32 (Figure 1A). This allows SAP to simultaneously bind SLAM and Fyn to form a trimeric complex [26,27]. Fyn contributes both to the phosphorylation of SLAM and to the recruitment of further signalling intermediates leading to the activation of downstream pathways essential for SLAM receptor functions [28,29,30]. In vivo experiments on both Fyn-silenced and SAP-silenced mice demonstrated that both NK-mediated cytotoxicity and cytokine production were compromised, suggesting that the association between the two proteins is required for NK-cell activation [31]. 

More than 50 mutations affecting the SH2D1A gene have been identified in XLP-1 patients, including micro/macro deletions, splice mutations, nonsense mutations and missense mutations, resulting in either partial or complete loss of the genetic product. Several missense mutations have been observed and characterized in vivo and in vitro to assess the mutants’ stability as well as their ability in binding SLAM [32]. Therefore, these mutations can be classified according to their ability to influence protein stability (by decreasing the half-life), or on their ability to impair the binding to the target proteins. Single amino acid substitution such as Y_7_C, V_10_G, S_28_R, Q_99_P, P_101_L and X_128_R in the SAP SH2 domain (Figure 1B) disrupt hydrophilic bonding resulting in a shorter half-life of the protein. Similarly, the substitution of the proline at the C-terminus of the SH2 domain with a leucine (P_101_L) is responsible for a different protein folding, probably resulting in a less stable product. Conversely, amino acid substitutions like R_32_Q and C_42_W (Figure 1B) selectively obstruct the SH2 phosphotyrosine binding pocket, resulting in a lower binding efficiency both to the phosphorylated and unphosphorylated form of SLAM [32]. Indeed, the Y_32_ residue is also highly conserved in other proteins containing the SH2 domain and it is tightly required for phosphotyrosine recognition and binding. Two other important substitutions are T_53_I and T_68_I (Figure 1B) which compromise SAP binding abilities to the unphosphorylated form of SLAM, without affecting its half-life. T_68_ is localized close to amino acids involved in residue +3 binding and its substitution with isoleucine drastically affects binding to SLAM [24].

## 3. Signalling Defects in XLP-1 and Their Biological Effects 

The ability of SAP to bind to specific ITMS present in the cytoplasmic portion of SLAM family proteins couples the Src-family kinase Fyn to the SLAM receptors through non-canonical surface-surface interactions between SAP SH2 domain and Fyn SH3 domain [27]. Thus, the activated Fyn kinase phosphorylates several downstream effectors such as Dok1, Dok2 and SH2 containing inositol phosphatase (SHIP), resulting in NFκB signalling activation and IFN- γ production [33,34]. Interestingly, Cannons et al., demonstrated that in CD4^+^ cells NFκB activation is mediated by PKCθ, which is recruited by the SAP/Fyn pathway (Figure 2). Consequently, in SAP or Fyn deficient conditions this signalling is strongly impaired and IL-4 production is compromised [30] (Figure 3A). In the following studies, by pull-downs and co-immunoprecipitation assays, they demonstrated that SAP constitutively binds to PKCθ independently by Fyn interaction and that this SAP- PKCθ association triggers a signal transduction which leads to IL-4 expression upon T-cell stimulation [35]. In particular, the obtained data indicates that R_78_ in SAP sequence plays a key role in the formation of this complex, and interestingly, it has been shown that this residue also mediates an association with the SH3 domain of PAK-interacting exchange factor (PIX), promoting the formation of a trimeric complex SAP-PIX-Cdc42 [36]. Furthermore, through a screening of regulatory proteins containing the SH3 domain, Li and colleagues identified NCK Adaptor Protein 1 (NCK1) as a novel SAP binding partner, observing that the knock-down of SAP reduced phosphorylation of NCK1 and other proteins downstream to TCR signalling, including LAT and SLP-76, resulting in decreased cell proliferation and ERK pathway activation [37].

However, SAP not only acts as an adaptor protein, but also as a key regulator for the balance between activating and inhibitory signals downstream SLAM receptors, as it has been demonstrated that its binding ITSM sequences prevent the recruitment of both SHIP and SHP-2 by competition, blocking the phosphatase’s inhibitory functions [34,38] (Figure 2). Interestingly, it has been shown that, in SAP-deficient conditions, the interaction between SHP-1/SHP-2 with SLAM receptors may have a role in reducing TCR signalling intensity [31,39]. Furthermore, SAP has been identified as a direct binding partner of the ITAM sequences of the CD3ζ chain in the proximal portion of the cell membrane. In line with this, studies by Proust and colleagues indicates that SAP silencing reduces various TCR signalling pathways, such as ERK, Akt and PLCγ1 and decreases both IL-2 and IL-4 mRNA production [40] (Figure 3A). Interestingly, it has been observed that SAP can also interact with some adhesion molecules such as platelet–endothelial cell adhesion molecule-1 (PECAM-1) through the phosphorylated tyrosine 686, mediating adhesion processes in T cells. CD3ζ and PECAM-1 binding suggest that SAP is also involved in other signalling pathways not implicated in SLAM-receptor functions [41]. 

In line with reduced signalling, in absence of SAP, CTLs also show an impaired restimulation-induced cell death (RICD), a particular kind of apoptosis that constitutes an autoregulatory mechanism to prevent excessive lymphoproliferation and maintain cell homeostasis (Figure 3A). Results obtained by Snow et al. strongly indicate that T cell restimulation triggers apoptosis only when a certain threshold is reached. The authors suggest that SAP may act as a signal amplifier that increases the TCR strength up to the threshold required for RICD, resulting in an increased expression of pro-apoptotic molecules such as FASL and BIM. Conversely, in SAP-deficient conditions, the TCR signal strength is attenuated below the threshold, allowing T-lymphocytes to escape cell death [42] (Figure 3A). Notably, it has been shown that upon TCR stimulation, SAP mediates the association of LCK with the SLAM family receptor NK, T, and B Ag (NTB-A), enhancing pro-apoptotic signals in human T-cells. This signalling complex amplifies the TCR signalling, thereby promoting RICD in SAP dependent manner. Conversely, in SAP deficient conditions, the association between LCK and NTB-A is impaired resulting in reduced TCR strength contributing to RICD resistant cell phenotype [43]. In addition, it has been demonstrated that SAP can also associate with inhibitory T-cell receptors, such as programmed cell death-1 (PD-1). Through affinity purification and mass spectrometry analysis, Peled and colleagues demonstrated that SAP counteracts PD-1 inhibitory functions in T cells, blocking SHP-2 activation [44]. Thus, in the following study conducted on rheumatoid arthritis patients, the same authors confirm that SAP exerts an inhibitory effect on the PD-1 signalling pathway preventing SHP-2 dephosphorylation on tyrosine 173 of the CD28 receptor. The authors propose a mechanism by which rheumatoid arthritis T cells increase SAP expression or block its degradation in order to avoid PD-1 mediated exhaustion [45].

In the XLP syndrome context, SAP mutations are mostly loss of function and the biological effects derived from this immune defect have been investigated in recent years. Cells derived by SAP-deficient murine models exhibit functional dysregulations in both CD4^+^ and CD8^+^ T cells, impaired IL-4 IL-13 and IL-10 cytokines production, altered germinal-centre formation and a reduced response to TCR mediated cell activation [46]. Furthermore, XLP patients are characterized by a reduced number of CD4^+^ T cells displaying a T helper 2 phenotypes, as well as by a loss of circulating memory B cells and iNKT cells [47,48]. SAP-deficient CD8^+^ T cells show specific defects in their cytotoxic activity [49,50,51], while CD4^+^ have reduced activity in supporting B cell maturation [52,53]. Those defects are due to impaired/transient immunological synapse (IS) formation and an inefficient actin clearance from the IS central region. Indeed, Zhao et al., demonstrated that SAP is essentially required for effective T-B cell interaction and killing of EBV infected B cell targets. The authors provide evidence that in absence of SAP, the association between SHP-1 and SLAM receptors increases, promoting a negative signal which results in decreased activation of Src family kinases and tyrosine phosphorylation (Figure 3A), both crucial events in early IS formation. Conversely, in the presence of SAP, TCR-mediated positive signals are enhanced and CTL functions are more effective (Figure 2). These data highlight the essential role of SAP and SLAM family proteins in regulating positive and negative signals required for IS organization and CTLs lytic function [48]. 

Summarizing, SAP is a key multi-functional protein exerting specific roles in different signalling pathways downstream the TCR/SLAM receptors protein family. In absence of SAP, signalling is severely perturbed, impairing T cell and NK homeostasis and function.

## 4. SAP Controls DGK Activity and DAG Dependent Signalling

Triggering of the TCR promotes a rapid and sustained translocation of DGKα and DGKζ to the plasma membrane [54,55,56]. While DGKζ is distributed to all of the cell membrane, DGKα is driven at the periphery of the immune synapse [57] by the combined action of calcium binding to the EF-hands motifs, phosphorylation of tyrosine 335 by Lck [54,58] and a still partially characterized interaction with phosphatidylinositol 3 kinase (PI3K) products [57,59]. Despite its abundance and membrane recruitment, DGKα plays a quantitatively minor role in DAG metabolism in T cells [60,61] suggesting the existence of some mechanism to finely tune its activity and restricting it at the periphery of IS and some intracellular membranes [57,62]. Indeed, we and others have observed that DGKα but not DGKζ is inhibited upon T cell triggering with strong TCR agonist antibodies or upon TCR/CD28 co-stimulation. This DGKα inhibition is required to obtain a full T cell response [2,15,63]. The molecular mechanism involved in negatively regulating DGKα activity is still unknown but our observations indicated that it requires PLCγ activity and calcium, suggesting that this happens at the membrane in proximity of the receptor. Using inhibitors we excluded the involvement of either Src family kinases or PI3K activity [2]. Notably, this inhibition can be induced by triggering a recombinant SLAM receptor or by overexpressing SAP. This observation coupled with the finding that SAP is necessary for DGKα inhibition upon TCR/CD28 co-stimulation clearly indicates a central role of SAP in promoting DGKα inhibition (Figure 2) [2]. However, a direct SAP-DGKα interaction was not observed, as ITMS are not present on DGKα. The observation that SAP mutants R_55_L and R_76_A, defective in respectively phosphotyrosine binding and downstream effector recruitment, are not capable of DGKα inhibition, suggests that SAP acts by recruiting further unidentified mediators. The critical effector is not Fyn as Src family activity activates DGKα [64] and Src family inhibitor PP2 does not affect DGKα inhibition triggered by TCR/CD28, while the effect of PP2 on DGKα inhibition induced by SLAM receptors may be due to their intrinsic dependence on Fyn for signal transduction [2].

The observation that, in the absence of SAP, DGKα is predominantly active, suggests that the signalling defects observed in T cells from XLP-1 patients are in part due to DAG metabolism by DGKα and can be restored by DGKα inhibition (Figure 3A,B). Indeed, DGKα inhibitors or DGKα silencing restore DAG accumulation at the IS and the polarization of T cells toward B cell targets. They also rescue DAG dependent recruitment of PKCθ and RasGRP1, which drives Ras and the MAPK pathway along with promoting NFAT transcriptional activity and the expression of IL-2 and CD25 (Figure 3B). Intriguingly, the signalling rescue is not total, as BIM and Fas ligand expression remain downregulated in SAP deficient cells despite DGKα silencing/inhibition, indicating the existence of TCR signalling pathways that are SAP mediated but DGKα independent [2,16]. 

## 5. DGKα and DGKζ Inhibitors as Potential XLP-1 Therapies

The rescue of DAG signalling obtained by targeting DGKα in SAP deficient cells is sufficient to restore both cytotoxic activity towards B-cell targets and RICD (Figure 3B). The restoring of RICD requires PKCθ and MAPK pathway activity. Those pathways trigger the intrinsic apoptosis pathway by enhancing DAG-mediated induction of NUR77 (NR4A1) and NOR1 (NR4A3) and their phosphorylation via MAPK-regulated 90-kD ribosomal S6 kinase (RSK) (Figure 3B) [16]. Phosphorylated NUR77 is known to exit from the nucleus and promote transition pore opening [65], thus promoting apoptosis despite the lack of FAS ligand and BIM induction in SAP deficient lymphocytes.

Thus, targeting DGKα activity in SAP deficient cells emerged as a promising strategy for XLP-1 therapy (an unmet clinical need). However, pharmacological targeting of DGKα is quite challenging, as there is no detailed structural information on DGK catalytic domain allowing rational drug design to develop new inhibitors and or refine their specificity against a single isoform. Thus far, a few DGK inhibitors have been discovered, but their limitations, such as off-target effects, lack of selectivity, low potency and poor pharmacokinetic properties, limit their clinical use [66]. 

The two commercially available allosteric DGK inhibitors, 3-[2-[4-(bis(4-Fluorophenyl)methylene)-1-piperidinyl]ethyl]-2,3-dihydro-2-thioxo-4(1H)-quinazolinone (R59949) [67] and 6-(2-(4-[(4-fluorophenyl)phenylmethylene]-1-piperidinyl)ethyl)-7-methyl-5H-thiazolo(3,2-a)pyrimidin-5-one (R59022) [68] are widely used in vitro. Both these inhibitors can revert the RICD defects in SAP deficient lymphocytes [16]. Furthermore, R59022 also showed beneficial effects in an in vivo model of XLP-1 [16]. In LCMV infected SH2D1A^-/-^ mice, R59022 (2 mg/kg) significantly reduced the activated CD8^+^ T-cells number along with lowering liver lymphocytic infiltrates and reducing serum IFNγ levels [16]. However, along with their poor pharmacological properties, both R59022 and R59949 exhibits serotonin antagonism which makes their use in human patients unlikely [69]. 

Ritanserinis a well-known serotonin receptor antagonist with a strong structural similarity with both R59022 and R59949. Indeed, ritanserin and its chemical fragment, RLM001, were reported as DGK inhibitors with high selectivity towards α isoform [69,70]. Ritanserin is a very promising compound for drug repurposing or repositioning, as it showed utility in glioblastoma animal models and trough clinical trials for the treatment of psychic disorders like alcohol dependence and schizophrenia, and was proved to be safe for human use [71]. On the other hand, some interesting compounds, CU-3 (5-((2E)-3-(2-furyl)prop-2-enylidene)-3-[(phenylsulfonyl)amino]s-2-thioxo-1,3-thiazolidin-4-one) and compound A were identified which selectively inhibited DGKα with very low IC_50_ values [72,73]. Previous studies from our group showed that similar to R59′s, both ritansern and CU-3 rescued the RICD defects in the XLP-1 phenotype [16,74]. However, being a selective 5-HT2A and 5-HT2C serotonin receptor antagonist, ritanserin actions on the central nervous system may limit its use for XLP-1 therapy [69]. Similarly, the structure and high reactivity of CU-3 constrained its utility in in vivo applications [74]. 

Sequentially, through virtual screening, we identified Amb639752 (1-(2,6-dimethyl-1H-indol-3-yl)-2-[4-(furan-2-ylcarbonyl)piperazin-1-yl]ethenone) as a novel DGK inhibitor [74]. Structure-activity-relationship studies on Amb639752 resulted in compound 11 and 20 which are highly active in inhibiting DGKα [75]. Even though they have some 3D structural similarities to R59949, Amb639752 and its analogues do not affect serotonin receptors [74,75]. The high specificity towards DGK α isoform and no off-target properties against serotonin receptors makes this molecule an interesting tool in terms of DGKα inhibition. Moreover, Amb639752 and its analogues rescued the RICD defects in SAP deficient lymphocytes without affecting RICD sensitivity in control lymphocytes, indicating their potential utility in treating XLP-1 patients [74,75]. Similar to ritanserin and R59s, Amb639752 can promote pro-apoptotic activity through the induction of NUR 77 (NR4A1) and NOR1 (NR4A3), suggesting that this is the main pathway through which DGKα inhibition restores RICD defects in XLP-1 [16,74]. However, like other DGK inhibitors, Amb639752 and its daughter molecules have some drawbacks. Even though they have high efficiency in targeting DGKα, these molecules proved to be effective only in a micromolar (µM) range. In the absence of in vivo and clinical studies, their utility is limited to in vitro experiments. Nevertheless, the pharmacophoric model, resulting from those studies, may be of great help in identifying novel DGKα inhibitors that might be suitable for clinical use [76,77].

Notably, Ruffo et.al., reported that along with DGKα, silencing DGKζ also restores the RICD defects in SAP deficient lymphocytes even if DGKζ is not regulated by SAP [16]. Lack of specific DGKζ inhibitors never allowed us to test the hypothesis of pharmacological inhibition of DGKζ in rescuing RICD defects. Recently, substituted naphthyridinone derivatives have been patented as novel T cell activators that can inhibit both DGKα and DGKζ simultaneously with low toxicity in vitro, and in stability and bioavailability profiles [78]. Even though those patented molecules target both α and ζ DGK isoforms with very attractive IC_50_ values, the lack of in vivo and pharmacological studies constrains their utility in clinics. 

Conversely, other reports suggested the utility of SHP-2 inhibitors in XLP-1 conditions to restore the SAP deficiency defects. In XLP-1 patients where SAP is not expressed, the tyrosines on SLAM family members bind to several strong inhibitory molecules, especially SHP-1 and SHP-2 [31,48,79,80] which essentially block activation, development and function of T and NK cells [34]. Notably, in vitro targeting of SHP-1/SHP-2 rescued the cytolytic activity of SAP deficient cells against murine B-cells [48]. Several chemical small molecules have been identified as SHP-2 inhibitors in recent times and are reported to have positive effects in cancer cell lines in vitro. In particular, the SHP-2 inhibitors, SHP099 (100 mg/kg), and FGF401 (30 mg/kg) are used in vivo in FGFR-driven cancer mice models [81]. Furthermore, TNO155, RMC-4630, JAB-3312 and JAB-3068 are other available validated SHP-2 inhibitors that are in advanced clinical trials for the treatment of various cancers [76,77]. To our knowledge, none of those inhibitors was tested in XLP-1 models until now.

## 6. Discussion

The current treatments of choice for XLP-1 aim to contain the life-threatening development of EBV-induced HLH and the frequently arising lymphomas. Patients surviving HLH as well as asymptomatic ones are candidates for allogeneic hematopoietic stem cell transplant that has a consistent percentage of failures [82]. To face the unmet clinical needs of XLP-1 patients, several researchers aim for gene therapy/gene editing approaches that are still in the preclinical phase [83]. 

The idea of using small molecules to correct the signalling defects of SAP deficient cells was never translated in the clinic. Preclinical studies by our and other groups have developed DGKα, DGKζ and SHP-2 inhibitors that are, in principle, suitable for human use, but there are no registered trials with small molecules in XLP-1. This may be due to the incomplete knowledge of disease pathogenetic mechanisms but also the typical problems of translational research on rare diseases surely contribute. Indeed, XLP-1 patients are, by definition, rare and referred to several centres randomly. At diagnosis patients with HLH are typically very severe, requiring intensive treatment regimens followed by bone marrow transplant, while the treatment of asymptomatic brothers requires careful evaluation from an ethical point of view. Moreover, the efforts to develop patients’ registries to track disease natural history and eventually verify treatment efficacy are still really fragmented. 

It is interesting to speculate that the role played by DGKα activity in rising TCR signalling threshold and promoting RICD resistance may not be unique for XLP-1. Indeed, similar TCR signalling defects give rise to immunodeficiencies with or without EBV-induced HLH [84]. Some monogenic diseases are expected to present reduced TCR-induced DAG signalling such as Zap-70 deficiency, interleukin-2-inducible T-cell kinase (ITK) deficiency, X-linked immunodeficiency with magnesium defect (XMEN) or Wiskott Aldrich syndrome (WAS), and thus may be in principle compensated by DGKα targeting. The clarification of the signalling mechanisms that lead to inhibition of DGKα would allow this lipid kinase to be positioned better in the TCR signalosome and to identify the primary immunodeficiencies that apart from SAP are characterized by excessive DGKα activity and thus potentially targeted by the DGK inhibitors currently under development.

## Figures and Tables

**Figure 1 ijms-22-05816-f001:**
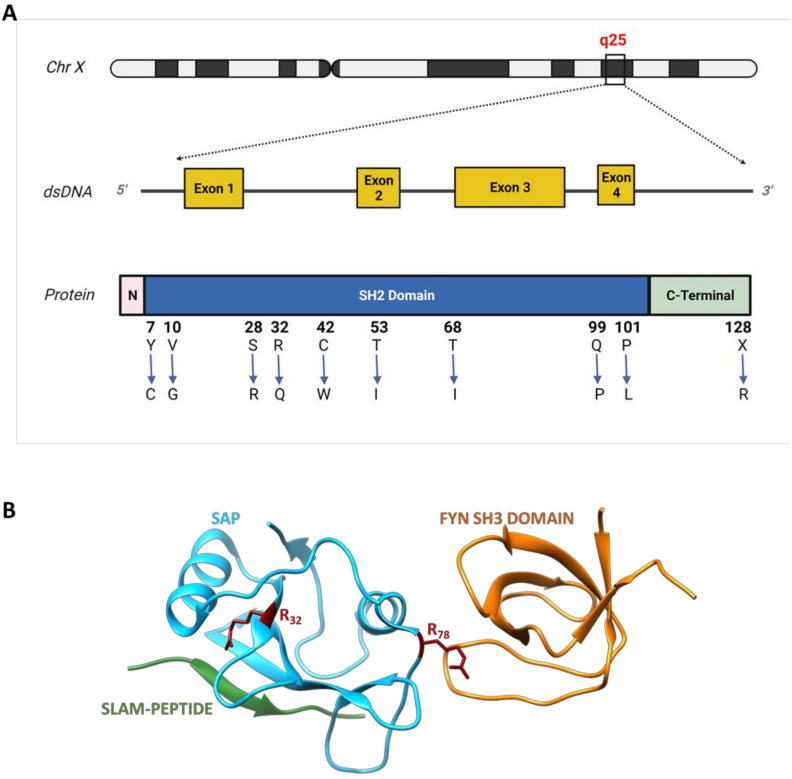
Schematic representation of SH2D1A gene with some missense mutations and the SAP protein structure. (**A**) The SAP gene and the corresponding protein are represented, showing an N-terminal (pink) region encoded by the exon 1, a central SH2 region (blue), encoded by exons 1,2 and 3 and a C-terminal region (light green) encoded both by exon 3 and 4 (exons are represented in yellow and indicated with a number in squares). Ten missense mutations on SAP SH2 and C-terminal domains affecting protein stability (Y_7_C, V_10_G, S_28_R, Q_99_P, P_101_L, X_128_R), and binding ability to both phosphorylated (R_32_Q, C_42_W) and unphosphorylated (R_32_Q, C_42_W, T_53_I, T_68_I) forms of SLAM are shown. (**B**) The 3D structure of SAP (light blue), in a trimeric complex with SLAM-PEPTIDE (green) and the SH3 domain of Fyn (orange) is showed [27]. The two arginine residues in positions 32 and 78, are shown (red) which are reported to play a critical role in the binding with SLAM and Fyn, respectively.

**Figure 2 ijms-22-05816-f002:**
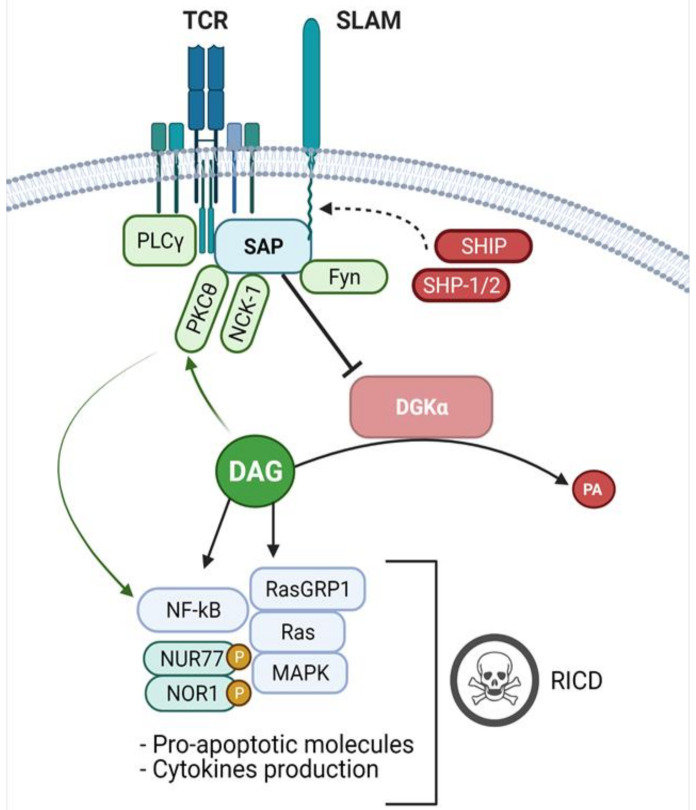
SAP signalling and DGKα inhibition. Upon TCR stimulation, the adaptor protein SAP binds to ITSMs sequences on the cytoplasmic tail of the SLAM-family and other receptors, competing with SHIP and SHP-1/2 phosphatases and promoting activator signals. At the membrane, SAP interacts with different binding partners, such as the FYN Kinase, PKCθ and NCK1, involved in T-cell activation. Moreover, SAP mediates the inhibition of DGKα, resulting in an accumulation of PLCγ-derived DAG, leading to MAPK pathway activation, pro-apoptotic molecules expression and cytokines production. Furthermore, the increased levels of DAG enhance PKCθ activity resulting in signalling which potentiates NFκB activation. In antigen-experienced CD8^+^ cells, these events trigger the RICD program, which promotes effector T-cells clearance and prevents excessive lymphoproliferation.

**Figure 3 ijms-22-05816-f003:**
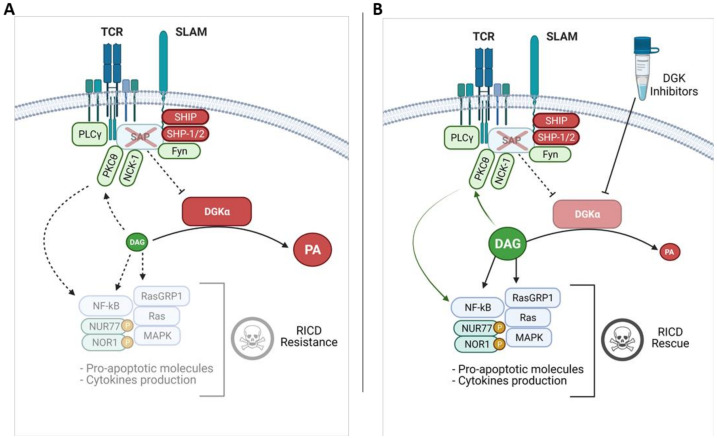
Reduced DAG signalling in XLP-1 and effects of DGK inhibitors. (**A**) In XLP-1 lymphocytes, the absence of SAP allows the binding of SHIP or SHP-1/2 phosphatases to SLAM family receptor ITSMs, promoting the transduction of inhibitory signals which attenuate the TCR signaling strength. In SAP-deficient conditions, DGKα shows increased activity and metabolizes DAG, contributing to RICD resistance. (**B**) This defect can be rescued by treating the cells with DGKα inhibitors which restore DAG signalling. DAG mediated PKCθ and MAPK activation leads to NUR77/NOR1 expression and phosphorylation, promoting RICD.

## Data Availability

Data sharing not applicable.

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
