# Peer review of "Diacylglycerol Kinase alpha in X Linked Lymphoproliferative Disease Type 1"

_ijms, 2021, doi:10.3390/ijms22115816_

Round 1

Reviewer 1 Report

Comments (ijms-1238335)

The review manuscript described by Valnati et al. focuses on XLP-1, SAP and DGKα. Although this review is concise, it is well written and provides new insights into XLP-1 therapy and DGK functions.

Minor point

Fig. 1: T52I in Fig. 2 vs.T53I in its legends and the text. Which is correct?

Author Response

The review manuscript described by Valnati et al. focuses on XLP-1, SAP and DGKα. Although this review is concise, it is well written and provides new insights into XLP-1 therapy and DGK functions.

We thanks the reviewer for appreciating our effort in providing a concise but stimulating state of art paper.

Minor point

Fig. 1: T52I in Fig. 2 vs.T53I in its legends and the text. Which is correct?

Corrected to T53I in figure.

Reviewer 2 Report

In this manuscript Velnati and coauthors performed an extensive revision of the molecular defects presented in the X-linked proliferative disease type I (XLP-1), including the alterations in DGKalpha activity. The signaling events that occurs in XLP-1 is a complex matter, and then an integrative review is helpful to improve the comprehension of the signaling landscape of this disease as well as its connection with other physiological events downstream TCR activation. The authors also put in context the XLP-1-signaling alterations with possible and potential pharmacological interventions that could mitigate this rare disease, including the blockade of DGKalpha and that of other molecules, like the phosphatases SHP-1, -2. Moreover, the authors call our attention to how the knowledge derived from XLP-1 could be applied to treat other diseases, which added relevance to this review. This in general makes the idea of this review valuable.  

Although in general the manuscript is well organized, there are some issues that should be addressed to facilitate its comprehension. The sections regarding SAP and DGKa inactivation, and the specificity of DGK inhibitors request some editing. These two sections are the essential parts of the review, and then their messages should be clearer.  There are some typos and in some paragraphs some connectors are used incorrectly or are used several times, a minor revision of should be performed to improve the quality.

Comments and suggestions are listed below:

L 34. ..but also Ras. Please consider rephrase by..  and also that of Ras.

L 40. ..but also . Please consider rephrase by and also of .

L 55.  …DGKz KO. Change KO by deficient.

L 65, 88 Even though, replace by other connector.

L 72. EBV is not peculiar… It is not clear if all the clinical manifestations are always due to EVB or could be triggered by other factors.

L 62 Section 2.1

For improve clarity, please consider introduce sooner that the SH2D1A gen is located in the long arm of the Chromosome X… (L177). This info could be moved in the paragraphs of the L74 or of L 83.

L 85 It is not clear if the ITSM motifs are in SAP or in SLAM. Although some information is added later (L97) it is not clear in the beginning. Please rephase for clarity.

Figure 1: This figure is an excellent idea to recap the information, however 1) it is confusing that the mRNA has the protein regions and the key aa. Please consider change mRNA by protein; 2) It could be easier to understand if the same code is used for described the aa in A than in B (R and arg), please consider the unification of the same aa code as it is used along the paragraph beginning in L117; 3) The use of very similar colors to depict SLAM and SAP makes difficult to understand the figure.

L 138. Please consider rephrase this subtitle, since in this section the functions of SAP in T cells are first described and then the alteration in XLP-1.

L 203. ..SAP contrast PD-1 inhibitory…   Is here “contrast” correct? Or it should say instead “counteracts”?

L 238…  partially uncharacterized…  consider change by partially characterized or not totally characterized.

L 233 Paragraph 2.3

This paragraph aims to review the data regarding the control of DGKa activity and SAP and is the central point in the manuscript. It is difficult to understand if DGKa is phosphorylated or not by a scr-kinase family to be inactivated, are different pools of the kinase involved? Compare L 237 vs 247. Please consider expand and/or reorder this paragraph.

Which is the reference stating that DGKa is inactivated by CD3/CD28? Is it 2? Please correct.

The next reference in this paragraph, referring to DGK activation by LCK is that referring to cancer cells and a growth factor. In my opinion here the reference 64 should be also included. Please correct.

L 262.  Consider change rescue by restore.

L 274. For improve clarity consider move the paragraph that starts at L265 here (Intriguingly, …). As written, it is difficult to read starting by this contrast of ideas.

L 292 consider eliminate “which”, it makes no sense in the phrase. Invitro, with itallics and spaced? Please check.

L297 Is “proprieties” correctly written? Or is it “properties”? Please check.

L 297-298. ..exhibit serotonin antagonism…

This idea is finished here but then in the next paragraph the use of ritanserin is described. This drug share structure with the R59´s, but this is not discussed. Reorganization of this two sections to mention the common structure and the targeting of serotonin of the three drugs could facilitate the reading.

L 315. SAR studies…, is SAR abbreviation annotated somewhere else?

L 325, L 329 Consider eliminate or replace one “However”

L 329 ..realized.. I do not understand the meaning of this word here.

L 336…  promising toxicity… Consider rephasing (reduced?) or explain why toxicity could be promising.

L 347  It is difficult to compare the different doses used in mouse of all these inhibitors (R59´s 2mg/day) vs 100 mg/kg or 30 mg/kg. 

L 348 … used in-vivo in.. in itallics? Please homogenize this in all the Ms.

L 375… some of those monogenic diseases…please eliminate “of those” of rephrase.

Author Response

Comments and Suggestions for Authors

In this manuscript Velnati and coauthors performed an extensive revision of the molecular defects presented in the X-linked proliferative disease type I (XLP-1), including the alterations in DGKalpha activity. The signaling events that occurs in XLP-1 is a complex matter, and then an integrative review is helpful to improve the comprehension of the signaling landscape of this disease as well as its connection with other physiological events downstream TCR activation. The authors also put in context the XLP-1-signaling alterations with possible and potential pharmacological interventions that could mitigate this rare disease, including the blockade of DGKalpha and that of other molecules, like the phosphatases SHP-1, -2. Moreover, the authors call our attention to how the knowledge derived from XLP-1 could be applied to treat other diseases, which added relevance to this review. This in general makes the idea of this review valuable.  

Although in general the manuscript is well organized, there are some issues that should be addressed to facilitate its comprehension. The sections regarding SAP and DGKa inactivation, and the specificity of DGK inhibitors request some editing. These two sections are the essential parts of the review, and then their messages should be clearer.  There are some typos and in some paragraphs some connectors are used incorrectly or are used several times, a minor revision of should be performed to improve the quality.

We thank the reviewer for appreciating our effort in reviewing the state of art in this field. We have also to thank the reviewer for the precise and appropriate comments. We revised the paper accordingly.

Comments and suggestions are listed below:

L 34. ..but also Ras. Please consider rephrase by..  and also that of Ras.

This sentence was rephrased to improve clarity.

L 40. ..but also . Please consider rephrase by and also of .

Rephrased accordingly.

L 55.  …DGKz KO. Change KO by deficient.

Rephrased accordingly.

L 65, 88 Even though, replace by other connector.

Rephrased accordingly.

L 72. EBV is not peculiar… It is not clear if all the clinical manifestations are always due to EVB or could be triggered by other factors.

We revised the paragraph to improve clarity.

L 62 Section 2.1

For improve clarity, please consider introduce sooner that the SH2D1A gen is located in the long arm of the Chromosome X… (L177). This info could be moved in the paragraphs of the L74 or of L 83.

Moved accordingly.

L 85 It is not clear if the ITSM motifs are in SAP or in SLAM. Although some information is added later (L97) it is not clear in the beginning. Please rephase for clarity.

Rephrased accordingly.

Figure 1: This figure is an excellent idea to recap the information, however 1) it is confusing that the mRNA has the protein regions and the key aa. Please consider change mRNA by protein; 2) It could be easier to understand if the same code is used for described the aa in A than in B (R and arg), please consider the unification of the same aa code as it is used along the paragraph beginning in L117; 3) The use of very similar colors to depict SLAM and SAP makes difficult to understand the figure.

Figure was modified to meet reviewer indications.

L 138. Please consider rephrase this subtitle, since in this section the functions of SAP in T cells are first described and then the alteration in XLP-1.

Rephrased “Signalling defects in XLP-1 and their biological effects.

L 203. ..SAP contrast PD-1 inhibitory…   Is here “contrast” correct? Or it should say instead “counteracts”?

Rephrased accordingly.

L 238…  partially uncharacterized…  consider change by partially characterized or not totally characterized.

Rephrased to “partially characterized” as suggested.

L 233 Paragraph 2.3

This paragraph aims to review the data regarding the control of DGKa activity and SAP and is the central point in the manuscript. It is difficult to understand if DGKa is phosphorylated or not by a scr-kinase family to be inactivated, are different pools of the kinase involved? Compare L 237 vs 247. Please consider expand and/or reorder this paragraph.

We improved the discussion of this point and detailed the reasons that suggest us Fyn not to be involved in DGKa inhibition.

Which is the reference stating that DGKa is inactivated by CD3/CD28? Is it 2? Please correct.

Reference 2 inserted.

The next reference in this paragraph, referring to DGK activation by LCK is that referring to cancer cells and a growth factor. In my opinion here the reference 64 should be also included. Please correct.

Reference 64 inserted.

L 262.  Consider change rescue by restore.

Rephrased accordingly.

L 274. For improve clarity consider move the paragraph that starts at L265 here (Intriguingly, …). As written, it is difficult to read starting by this contrast of ideas.

We modified L 274 to make it clearer.

L 292 consider eliminate “which”, it makes no sense in the phrase. Invitro, with itallics and spaced? Please check.

Corrected.

L297 Is “proprieties” correctly written? Or is it “properties”? Please check.

Corrected

L 297-298. ..exhibit serotonin antagonism…

This idea is finished here but then in the next paragraph the use of ritanserin is described. This drug share structure with the R59´s, but this is not discussed. Reorganization of this two sections to mention the common structure and the targeting of serotonin of the three drugs could facilitate the reading.

We highlighted the structural homology between ritanserin and R59s in the text. A clear statement about action of those molecules on the serotoninergic system is already present in the text and we prefer not to duplicate those comments.

L 315. SAR studies…, is SAR abbreviation annotated somewhere else?

Abbreviation removed.

L 325, L 329 Consider eliminate or replace one “However”

The second one was replaced with “Nevertheless”.

L 329 ..realized.. I do not understand the meaning of this word here.

Rephrased.

L 336…  promising toxicity… Consider rephasing (reduced?) or explain why toxicity could be promising.

Changed to “low toxicity in vitro”.

L 347  It is difficult to compare the different doses used in mouse of all these inhibitors (R59´s 2mg/day) vs 100 mg/kg or 30 mg/kg. 

We verified the units of R59022 and changed to 2 mg/kg as it is reported.

L 348 … used in-vivo in.. in itallics? Please homogenize this in all the Ms.

Done.

L 375… some of those monogenic diseases…please eliminate “of those” of rephrase.

Eliminated.